# Tensor network study of the Shastry-Sutherland model with weak interlayer coupling

Patrick C. G. Vlaar[1*] and Philippe Corboz[1†]

**1** Institute for Theoretical Physics and Delta Institute for Theoretical Physics,
University of Amsterdam, Science Park 904, 1098 XH Amsterdam, The Netherlands
* p.c.g.vlaar@uva.nl     † p.r.corboz@uva.nl

June 9, 2023

## Abstract

**The layered material $SrCu_2(BO_3)_2$ has long been studied because of its fascinating physics in a magnetic field and under pressure. Many of its properties are remarkably well described by the Shastry-Sutherland model (SSM) - a two-dimensional frustrated spin system. However, the extent of the intermediate plaquette phase discovered in $SrCu_2(BO_3)_2$ under pressure is significantly smaller than predicted in theory, which is likely due to the weak interlayer coupling that is present in the material but neglected in the model. Using state-of-the-art tensor network methods we study the SSM with a weak interlayer coupling and show that the intermediate plaquette phase is destabilized already at a smaller value around $J''/J \sim 0.04 - 0.05$ than previously predicted from series expansion. Based on our phase diagram we estimate the effective interlayer coupling in $SrCu_2(BO_3)_2$ to be around $J''/J \sim 0.027$ at ambient pressure.**

## 1   Introduction

The competing interactions in frustrated materials give rise to a rich variety of fascinating phenomena. A paradigmatic example is the layered material $SrCu_2(BO_3)_2$ which has attracted significant attention in the past decades, in particular since the discovery of its intriguing sequence of magnetization plateaus at 1/8, 2/15, 1/6, 1/4, 1/3, and 1/2 (and possibly 2/5) [1–11]. Substantial efforts have been invested in understanding the magnetic structures of the plateaus, with a growing consensus that they correspond to crystals of triplets at high magnetic field [12–18] and crystals of bound states of triplets at low field [19, 20]. Another exciting direction has been the study of the phase diagram under pressure [10, 21–29], which has revealed two phase transitions at zero field, including a critical point at finite temperature [27], and an even richer phase diagram at finite field [28].

Many properties of $SrCu_2(BO_3)_2$ are remarkably well described by the Shastry-Sutherland model (SSM) [1, 12, 30], a frustrated S=1/2 spin model of orthogonal dimers on a square lattice, shown in Fig. 1(a). Its Hamiltonian is defined as

$$\mathcal{H}_{2D} = J \sum_{\langle i,j \rangle} \mathbf{S}_i \mathbf{S}_j + J' \sum_{\langle i,j \rangle'} \mathbf{S}_i \mathbf{S}_j, \tag{1}$$

with $J$ and $J'$ the intra- and interdimer coupling, respectively. For small values of $J'/J$ the ground state is exactly given by a product of dimer singlets. In the other limit, the model reduces to the square lattice Heisenberg model with an antiferromagnetic (Néel) ground state. For intermediate $J'/J$ consensus has been reached on the existence of an empty plaquette (EP) state [31–34] in the range $0.675(2) < J'/J < 0.765(15)$ [34], in which strong bonds are

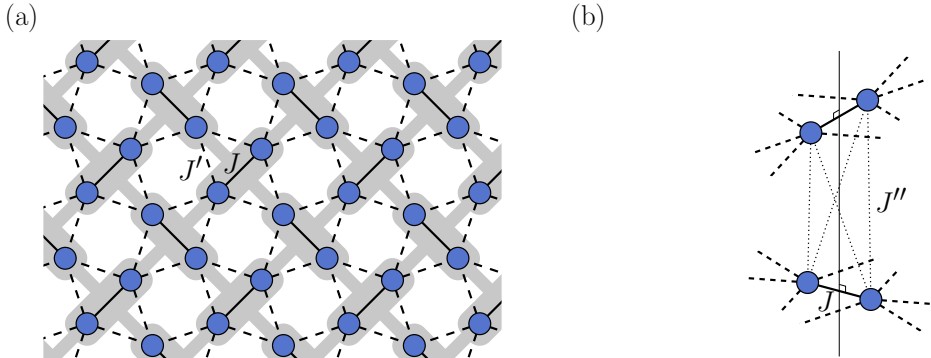

Figure 1: (a) The Shastry-Sutherland lattice with $J$ the intradimer and $J'$ the inter-dimer coupling. In gray the underlying square lattice of orthogonal dimers is shown. (b) Two dimers in adjacent layers interact via the interlayer coupling $J''$ as indicated by the dotted lines. Each site of a dimer is coupled to both sites of the orthogonal dimer in the neighboring layer.

formed around half the empty plaquettes which do not contain a dimer. While a weak first order phase transition between the EP and Néel phase was found in Ref. [34], more recently there have also been predictions of a deconfined quantum critical point [35] (see also related Ref. [29]) or a narrow quantum spin liquid region between the two phases [36, 37].

At ambient pressure the effective coupling ratio in $SrCu_2(BO_3)_2$ is around $J'/J = 0.63$ [9], which lies in the dimer phase but close to the plaquette phase. Applying hydrostatic pressure causes the Cu-O-Cu angle to diminish which results in a rapid decrease of $J$ and a slower decrease in $J'$ [38, 39], such that the ratio $J'/J$ increases. Evidence of a phase transition into an intermediate gapped phase around 1.8 GPa has been found in various experiments, including NMR [21], X-ray scattering [22], inelastic neutron scattering (INS) [23], electron spin resonance (ESR) [24], magnetization measurements [10, 28], and specific heat measurements [26, 27]. The nature of the intermediate phase is still not fully settled. There exist indications from NMR [21, 29] and INS [23] that the intermediate phase is not the EP phase but a closely related full plaquette (FP) phase, in which strong bonds are formed around the plaquettes containing a dimer. In the SSM this state is slightly higher in energy than the EP state, but it can be stabilized in a weakly distorted SSM [40] (however, the strength of the relevant couplings is not known). Antiferromagnetic order was observed with INS below 117 K and above 4 GPa [23], close to a tetragonal-monoclinic transition beyond which the SSM is no longer valid. Based on specific heat measurements [26], another Néel phase with a substantially lower transition temperature was discovered below 4 GPa, and the transition to the plaquette phase was found to occur around 2.5-3 GPa. Converted to $J'/J$ [28] this corresponds to $J'/J \sim 0.7 - 0.71$, which is significantly lower than the theoretical prediction based on the SSM.

A natural cause of this discrepancy is the presence of small interlayer couplings in the compound [26, 41], which are neglected in the 2D model. The dominant interaction can be described by an additional Heisenberg term with strength $J''$ between the layers [15, 41, 42], such that the 3D model reads

$$\mathcal{H}_{3D} = \mathcal{H}_{2D} + J'' \sum_{\langle i,j \rangle''} \mathbf{S}_i \mathbf{S}_j. \tag{2}$$

The $CuBO_3$ layers are stacked in such a way that the dimers have an alternating orientation in neighboring layers [15] and each site of a dimer interacts with both sites of the neighboring

dimer, see Fig. 1(b).[1] This model has already been studied in Ref. [41] using series expansion (SE), where it was found, based on a fourth order expansion, that the extent of the plaquette phase shrinks rapidly with increasing $J''$, and that it disappears beyond $J''/J \sim 0.08$. However, results at higher orders or from other numerical approaches have so far been lacking.

Regarding the strength of the interlayer coupling in $SrCu_2(BO_3)_2$, there is still no consensus. In Ref. [43] an estimate of $J''/J = 0.09$ was obtained from fits to the magnetic susceptibility. In Ref. [44], based on an analysis of the bound state energies of the two-triplet excitations, a much larger value $J''/J = 0.21$ was found. Calculations from density-functional theory [38] predicted a much smaller ratio, $J''/J \leq 0.025$, but with values for $J$ and $J'$ which deviate considerably from other predictions.

In this paper we refine the phase diagram of the SSM with weak interlayer coupling using state-of-the-art tensor network (TN) methods for layered systems, the layered corner transfer matrix (LCTM) algorithm which was introduced recently [45]. It is based on a 3D version of the infinite projected-entangled pair state (iPEPS) [46–48], a variational tensor network ansatz for ground states in the thermodynamic limit, which has already proven to be a powerful tool to study the SSM in 2D [9,11,19,28,34,49] (and also at finite temperature [27,50,51]). We show that the EP phase becomes unstable at even smaller values of $J''/J$ than predicted by SE. From our phase diagram we extract an estimate of $J''/J$ in $SrCu_2(BO_3)_2$, by determining the value which leads to an extent of the plaquette phase that is consistent with experiments. Besides this, we also analyze the effect of the interlayer coupling on the competition between the EP phase and the FP phase and show that the latter remains higher in energy also for finite $J''/J$.

The paper is organized as follows. In Sec. 2 a brief introduction to iPEPS and the LCTM method is given, together with additional simulation details. In Sec. 3 we first provide an overview of the phase diagram, followed by a detailed study of the phase transitions along selected cuts. We then present additional results on the competition between EP and FP states and the phase diagram, and end the results section with a discussion on the estimate of the effective interlayer coupling in $SrCu_2(BO_3)$. Finally, in Sec. 4 we present our conclusions.

## 2   Methods

### 2.1   Infinite projected entangled-pair states and LCTM method

To simulate the SSM with interlayer coupling we make use of tensor network methods based on projected entangled-pair states (PEPS) [46, 47]. A PEPS is a variational wave function ansatz for two- or higher-dimensional ground states given by a trace over a product of tensors on a lattice,

$$|\psi\rangle = \sum_{s_1 \ldots s_N=1}^{d} \mathrm{Tr}\Big( T_{s_1}^{\vec{r}_1} \ldots T_{s_N}^{\vec{r}_N} \Big) |s_1 \ldots s_N\rangle, \tag{3}$$

with $T_{s_i}^{\vec{r}_i}$ a tensor at position $\vec{r}_i$ and $s_i$ representing the index of the local Hilbert space of dimension $d$, see Fig. 2(a). For the present model we consider a cubic lattice with one tensor per dimer [34]. Besides the physical index, each tensor has six virtual indices which connect to the nearest-neighbor tensors. The dimension of the virtual indices, called the bond dimension $D$, determines the amount of entanglement that can be captured by the ansatz, i.e., it

---

[1]We note that the model neglects the buckling of the $CuBO_3$ layers which causes a small difference in the distance between the $Cu^{2+}$-ions in adjacent layers [15], and hence slightly different $J''$ couplings. However, since the effect is expected to be small, we consider the same coupling on all bonds for simplicity, as previously done in Ref. [41].

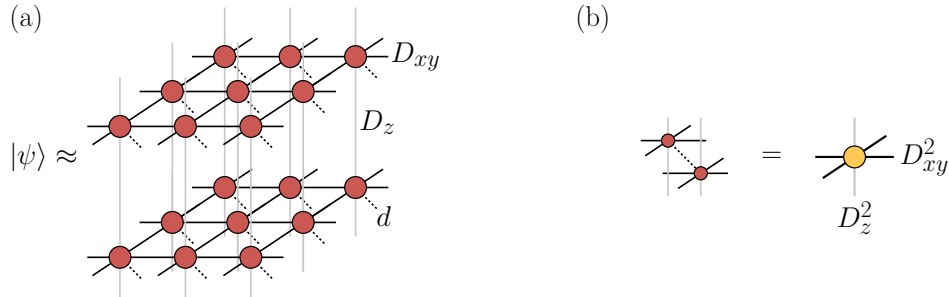

Figure 2: (a) Anisotropic iPEPS ansatz on a cubic lattice with intra- and interlayer bond dimensions $D_{xy}$ and $D_z$, respectively. The physical index is indicated by a dashed line. In (b) the norm tensor is displayed which represents the combined bra- and ket-iPEPS tensors on a site.

controls its accuracy. Motivated by the anisotropic nature of the model we take an ansatz with different bond dimensions in the intra- and interplane directions, $D_{xy}$ and $D_z$, respectively, with $D_{xy} \geq D_z$. An advantage of the PEPS ansatz is that it can directly describe states in the thermodynamic limit by defining a supercell of tensors and repeating it infinitely many times, which is called an infinite PEPS (iPEPS) [48]. In this work we consider a supercell consisting of two tensors which are repeated on the two sublattices. To improve the computational efficiency we use tensors with implemented $U(1)$ symmetry [52, 53].

A central element of the algorithm is the approximate contraction of the tensor network which is required, e.g., to evaluate expectation values of local observables. A commonly used method for iPEPS in 2D is the corner transfer matrix (CTM) approach [54–56], in which the contracted network is approximated by four corner and four edge tensors, each representing an infinite quadrant and infinite half-row of the network, respectively. The accuracy of the CTM method is systematically controlled by the environment bond dimension $\chi$. Tensor networks in 3D are significantly more challenging to contract than in 2D. Contraction methods for general cubic lattices do exist, such as HOTRG [57] or SU+CTM [58], but their computational cost is significantly higher than the 2D CTM approach. Here, we make use of the recently introduced layered CTM (LCTM) method [45], which is designed to efficiently contract iPEPS representing weakly-coupled layered systems.

The main ideas of the LCTM approach can be summarized as follows. Consider contracting the 3D network representing the norm shown in Fig. 3, which can also be used to evaluate an expectation value of a local operator (by placing an operator between the bra- and ket-iPEPS tensors in the center). First, a decoupling between the layers is performed away from the center by truncating the interlayer bonds from $D_z > 1$ to $D_z = 1$. This reduces the tensors away from the center to quasi-2D ones, which can be contracted using the 2D CTM method, resulting in the network shown in Fig. 3(b). In the center of each layer an untruncated tensor is kept with $D_z > 1$ to capture the most important correlations between the layers. Since the bonds in the z-direction carry only little entanglement, the truncation away from the center introduces only a small error on a local expectation value measured in the center. The remaining network consists of an infinite chain of tensors representing the layers connected in the center, which can be contracted by sandwiching the middle layer between the left and right dominant eigenvector of the transfer matrix represented by a contracted layer, as shown in Fig. 3(c). For details on the decoupling scheme and benchmark results, we refer to Ref. [45].

To obtain an iPEPS with optimal variational parameters to approximate the ground state, an optimization procedure has to be performed. Two commonly used techniques are imaginary time evolution and direct energy minimization. In the former approach an initial state

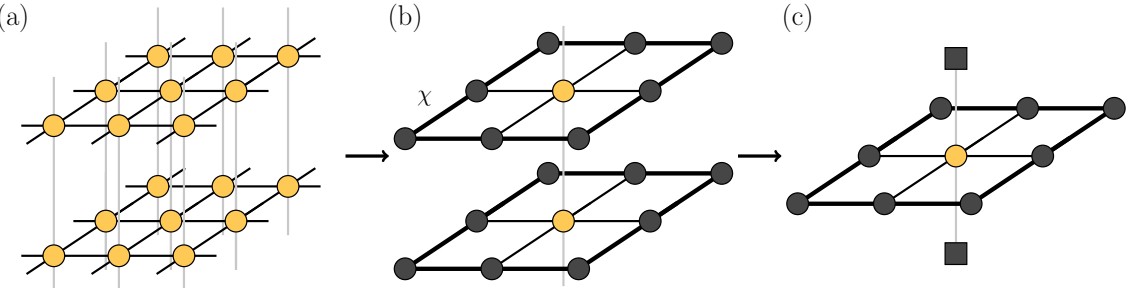

Figure 3: Main steps of the LCTM approach to contract the 3D tensor network representing the norm of an iPEPS shown in (a). Each norm tensor represents the combined bra- and ket-tensors on each site, as shown in Fig. 2(b). (b) The layers are decoupled away from the center by truncating $D_{\bar{z}} \to 1$ and the resulting quasi-2D tensor networks are contracted using the CTM method. (c) The resulting infinite 1D chain is contracted by replacing the neighboring layers by the corresponding left and right dominant eigenvector (black squares) of the transfer matrix represented by a contracted layer. A local expectation value in the center can be computed by inserting an operator between bra- and ket-tensors in the center and by dividing by the norm.

is projected onto the ground state by applying the imaginary time evolution operator $e^{-\beta\hat{H}}$, and taking $\beta \to \infty$ [48, 59]. A Trotter-Suzuki decomposition is used to split the imaginary time evolution operator into a product of two-body gates, which are then sequentially applied to the iPEPS. Applying a gate to a bond increases its bond dimension, which must be truncated to avoid an exponential growth. One way to do this is through the full update (FU) [48] approach in which the truncation is done by minimizing the norm distance $\left\| |\psi\rangle - |\psi'\rangle \right\|^2$ between the untruncated iPEPS $|\psi\rangle$ and the truncated iPEPS $|\psi'\rangle$. This requires the contraction of the TN environment around a bond which is computationally expensive. This cost can be significantly reduced by the fast-full update (FFU) [60] scheme which recycles the previous environment. This scheme can also be combined with the LCTM approach, see Ref. [45] for details. Alternatively to the imaginary time evolution the iPEPS tensors can also be optimized by minimizing the energy of the variational ansatz until convergence is reached [61–63].

## 2.2 Simulation details

For the present study we use a combination of energy minimization and FFU. Within the LCTM contraction approach we iterate the environment computation up to three times to obtain the interlayer projectors. Initially, the tensors are optimized on the 2D lattice (i.e., with $J''/J = 0$) by energy minimization. The resulting tensors are then used as initial states in a FFU optimization with finite $J''/J$, which we found leads to better results than optimizations initialized from a simple-update [64] optimization at the values of $J'/J$ and $J''/J$ considered. Simulations in the Néel phase are initialized at $J'/J = 0.8$ and evolved using imaginary time steps $\tau = 0.1$ and $\tau = 0.05$, where the state with lowest energy is selected. For the simulations of the EP phase, a 2D iPEPS at the corresponding $J'/J$ is used as initial state and the FFU is performed with $\tau = 0.05$. In some cases an increase in the energy is observed after a certain imaginary time $\beta$. In these cases the simulation is halted and the tensors giving the lowest energy are selected. We note that the EP states can be stacked in two different ways. While our main results have been obtained with plaquettes in alternating positions in adjacent layers, we have also tested the stacking with plaquettes on top of each other [35], which we found leads to similar results.

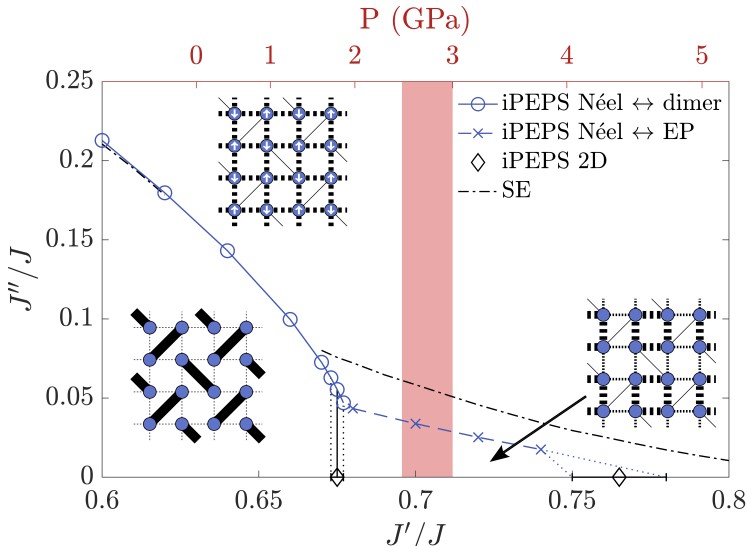

Figure 4: The phase diagram of the Shastry-Sutherland model with interlayer coupling ($D_{xy} = 6$, $D_z = 3$), which includes a dimer, a Néel, and an empty-plaquette (EP) phase. Previous iPEPS results for the 2D model [34] are shown by the black diamonds, and the dashed dotted lines correspond to fourth order series expansion data [41]. The upper horizontal axis shows the pressure corresponding to a particular coupling $J'/J$, based on the pressure model from Ref. [28]. The location of the plaquette to Néel phase transition found in experiments [26] is marked in light-red.

To identify the phases, we introduce the following order parameters. For the Néel phase we consider the local magnetic moment

$$m = \sqrt{\langle \mathbf{S}_x \rangle^2 + \langle \mathbf{S}_y \rangle^2 + \langle \mathbf{S}_z \rangle^2}, \tag{4}$$

with $\mathbf{S}_i$ the spin operators. As an order parameter for the EP phase we use

$$\Delta e_{\text{EP}} = \bar{e}_{\text{other}} - \bar{e}_{\text{EP}}, \tag{5}$$

where $\bar{e}_{\text{EP}}$ is the mean energy of the bonds belonging to the empty plaquette and $\bar{e}_{\text{other}}$ is the mean energy of the remaining bonds.

## 3   Simulation results

The main results of this work are summarized in the phase diagram in Fig. 4. For the range of couplings considered, the 3D model exhibits the same three phases as the 2D model: a dimer phase made of exact singlets in the low $J'/J$ region, an empty plaquette (EP) phase at intermediate values of $J'/J$, and a Néel phase which dominates for sufficiently large $J'/J$ and/or large $J''/J$. The EP phase destabilizes already at weak-interlayer coupling of at most $J''/J \sim 0.04 - 0.05$ in favor of the Néel phase, while the dimer phase survives at stronger interlayer coupling. That the Néel state gets energetically favored for sufficiently large $J''/J$ can be intuitively understood from the fact that the antiferromagnetic layers can be stacked on top of each other without frustrating the Néel order [41], and hence the additional interlayer coupling lowers the energy per site. This is in contrast to the interlayer energy of the 2D dimer which is exactly zero, or close to zero in case of the EP state.

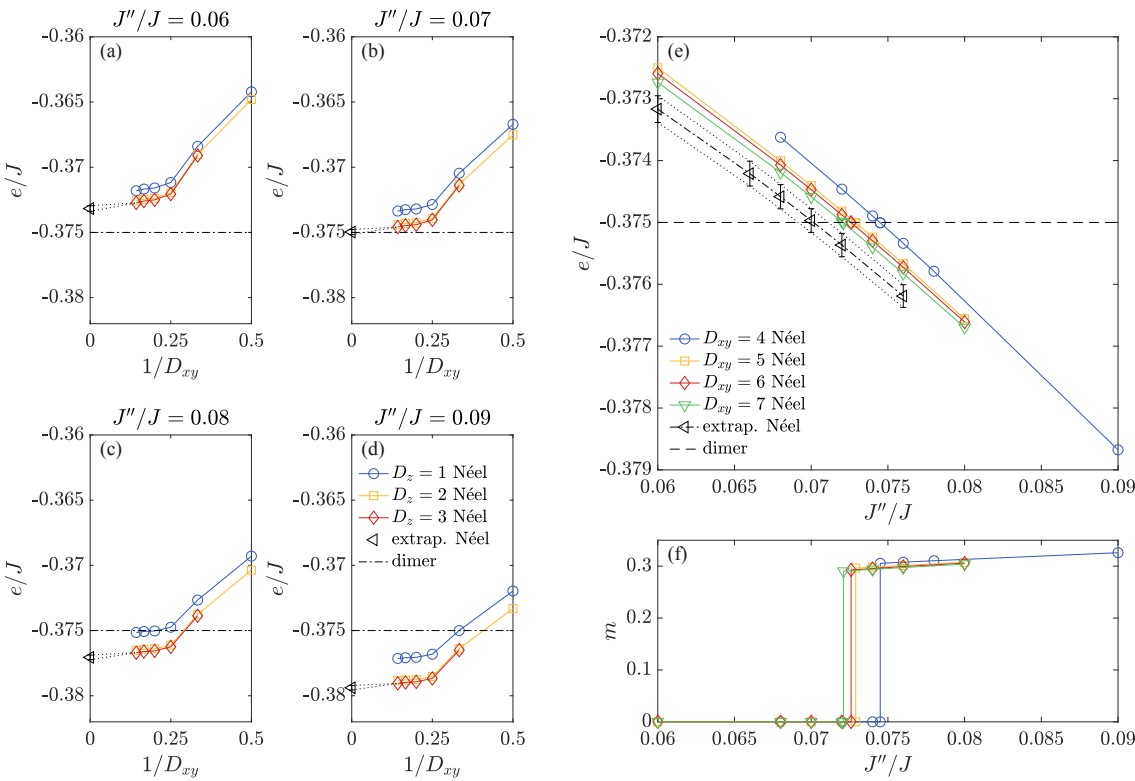

Figure 5: Results for the dimer to Néel phase transition for fixed $J'/J = 0.67$. (a)-(d) Energy per site $e$ in units of $J$ of the Néel state as a function of $1/D_{xy}$ for $D_z = 1-3$ and different values of $J''/J$, including an extrapolation to the infinite bond dimension limit (see text for details). The exact energy of the dimer state is shown by the dashed dotted line. (e) Energies of the dimer and Néel states as a function of $J''/J$ ($D_z = 3$), where the phase transition is located at the intersection of the energies. (f) Local magnetic moment $m$ across the phase transition, revealing that the transition is of first order.

Our results are in good agreement with the SE results from Ref. [41] for the dimer singlet to Néel phase transition, however, the transition from the EP to Néel phase is located at weaker interlayer coupling than predicted by SE shown by the dashed-dotted line in Fig. 4.

In the following we provide a detailed study of the phase transitions along specific cuts for fixed $J'/J$ and varying $J''/J$.

## 3.1 Phase transition between dimer and Néel phase

In this section we consider the phase transition between the dimer singlet and the Néel phase, focusing on a cut at $J'/J = 0.67$. We note that in Ref. [45] this transition has been studied before with the LCTM method for $J'/J = 0.61$ and $J'/J = 0.66$ as a benchmark example.

Figures 5(a)-(d) show the energy per site $e$ as a function of inverse bond dimension $1/D_{xy}$ and different values of $D_z$ at $J''/J = 0.06$, 0.07, 0.08, and 0.09, respectively. The dimer state has an exact energy of $e = -3/8J$ per site, i.e., it is independent of $J'/J$ and $J''/J$, and it is already reproduced for $D_{xy} = D_z = 1$. The energy of the Néel state exhibits a stronger dependence on $D_{xy}$ than on $D_z$, which is due to the weak interlayer coupling compared to the intraplane coupling. While the energy gets considerably lowered from $D_z = 1$ to $D_z = 2$, it hardly changes anymore when increasing it to $D_z = 3$. In contrast, substantially larger changes are obtained by varying $D_{xy}$, which motivates using an anisotropic ansatz with $D_{xy} > D_z$.

To obtain an estimate of the energy in the infinite bond dimension limit, we perform an extrapolation in $1/\kappa$, with $D_{xy} = \kappa$ and $D_z = \frac{\kappa-1}{2}$, i.e., using the energy of the states for $(D_{xy} = 5, D_z = 2)$, $(D_{xy} = 7, D_z = 3)$, and the mean of $(D_{xy} = 6, D_z = 2)$ and $(D_{xy} = 6, D_z = 3)$. Since the convergence in the bond dimension is typically faster than linear, we take the mean between the linearly extrapolated result and the lowest energy result at $(D_{xy} = 7, D_z = 3)$ to be the estimate. As an error estimate we take half the difference between the estimate and the lowest energy value at finite bond dimension.[2]

The location of the phase transition is obtained from the intersection of the energies of the two states as a function of $J''/J$, shown in Figure 5(e).[3] Since the energy of the dimer state is fixed, each finite bond dimension result for the critical coupling $(J''/J)_c$ corresponds to an upper bound. The largest bond dimension $(D_{xy} = 7, D_z = 3)$ yields $(J''/J)_c = 0.072$, whereas a slightly lower value is obtained based on the extrapolated energy, $(J''/J)_c = 0.0702(10)$.

In Fig. 5(f) the local magnetic moment $m$ of the lowest energy state at fixed $D$ is shown, revealing a large jump at the phase transition. Increasing $D_{xy}$ has only a small effect on $m$ near the phase transition which indicates that the jump remains finite in the infinite $D$ limit, corresponding to a first-order phase transition.

## 3.2  Phase transition between empty plaquette and Néel phase

We next consider the transition between the EP phase and the Néel phase, focusing on a cut at $J'/J = 0.7$. In Figs. 6(a)-(d) the energy of the states as a function of $1/D_{xy}$ is presented for $J''/J = 0.02, 0.03, 0.04$, and $0.05$ respectively, including an extrapolation performed in a similar way as described in the previous section.

The dependence of the Néel state energy on $D_z$ is weaker than in the case of $J'/J = 0.67$ in the previous section, due to the smaller interlayer couplings that are considered here. An even weaker dependence on $D_z$ is found for the EP state, which is expected due to the inherent 2D nature of the state. Since the local magnetic moment in the EP state vanishes, the interlayer energy is exactly zero at lowest order for $D_z = 1$, corresponding to a product state of 2D iPEPS. Going beyond $D_z = 1$ introduces correlations between the planes, which however remain very weak compared to the strong intraplane plaquette correlations, and hence the interlayer energy remains close to zero.

Figure 6(e) shows the energy of the states as a function of $J''/J$ in the vicinity of the phase transition. When $D_{xy}$ is increased from 4 to 6 the EP state decreases slightly faster in energy than the Néel state, causing the location of the phase transition to shift to slightly higher $J''/J$. When increasing $D_{xy}$ further only a tiny shift to a smaller coupling $(J''/J)_c = 0.034$ is found, suggesting that the location of the critical point does not change significantly anymore. Based on the extrapolated energies we get a critical coupling of $(J''/J)_c = 0.036$. By intersecting the upper and lower bounds of the error bars of the extrapolated energies we obtain an error range on the transition between $0.026 - 0.042$, which should provide a rather conservative error estimate. A less conservative error range is obtained by intersecting half the error bar widths, yielding $0.032 - 0.039$. Our value for the critical coupling is significantly lower than the SE result $(J''/J)_c = 0.058$ from Ref. [41].

In Figs. 6(f) and (g) the values of the local magnetic moment and EP order parameter across the phase transition are presented. Also in this case a discontinuous jump in the order parameters is found, indicating that the transition is of first order.

---

[2]We note that a more accurate energy extrapolation in the gapless Néel phase could be obtained using finite correlation length scaling [65, 66]. However, this requires an accurate estimate of the correlation length which with the current version of the LCTM approach cannot be obtained in a controlled way (since the approach is tailored to the computation of local observables).

[3]We note that, due to hysteresis effects, a state initialized in the Néel phase remains a Néel state even slightly beyond the transition into the dimer phase.

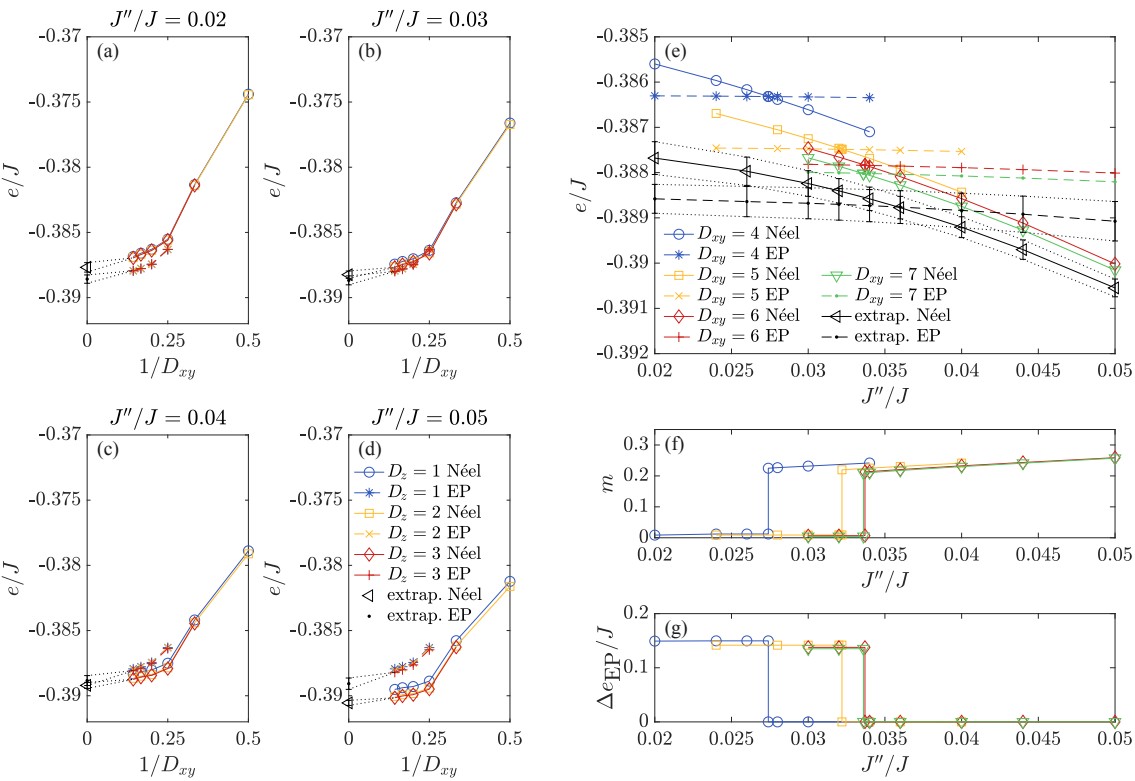

Figure 6: Results for the EP to Néel phase transition for fixed $J'/J = 0.7$. (a)-(d) Energy per site as a function of $1/D_{xy}$ for $D_z = 1-3$ and different values of $J''/J$, including extrapolated energies (see Sec. 3.1 for details on the extrapolation). (e) Energies of the EP and Néel states as a function of $J''/J$ ($D_z = 3$), where the critical coupling is located at the intersection of the energies. Both the local magnetic moment (f) and the EP order parameter (g) exhibit a jump at the phase transition, indicating that it is of first order.

## 3.3 Competition between the empty and full plaquette states

As mentioned in the introduction, the precise nature of the intermediate phase in $SrCu_2(BO_3)_2$ is still not fully confirmed. NMR [21,29] and INS [23] experiments suggest that the intermediate phase in $SrCu_2(BO_3)_2$ is not the EP but a full plaquette (FP) phase, in which strong bonds are formed around the plaquettes containing a dimer, as opposed to the EP state. In the SSM it was shown that the FP state is higher in energy than the EP state [40], but it can be stabilized by a relatively small deformation of the model using two types of intra- and interdimer couplings [40,67]. In the following we study the effect of the interlayer coupling on the competition between the EP and FP states at $J'/J = 0.7$, in order to test whether the FP state gets potentially stabilized in the 3D model.

In Figs. 7(a) and (b) we present the energies of the EP and FP states as a function of $J''/J$, respectively. The simulations for the FP state have been initialized in the 2D FP phase [40] by using two distinct dimer couplings on the two orthogonal dimers, $J_2/J_1 = 0.85$, and $J'/J_1 = 0.7$.[4] The dependence of the energy on $J''/J$ is found to be very weak for both states such that the FP state remains higher in energy than the EP state as in the $J''/J = 0$ case. For comparison, we have also added the energy of the purely 2D states by the dashed lines which lie close to the energy obtained from the FFU optimization with finite $J''$, showing that the

---

[4]Due to stability issues for $D_{xy} = 6$ with $\tau = 0.05$ we have used a smaller value of $\tau = 0.01$ for $D_{xy} = 6$ which, however, does not have a significant effect on the energy.

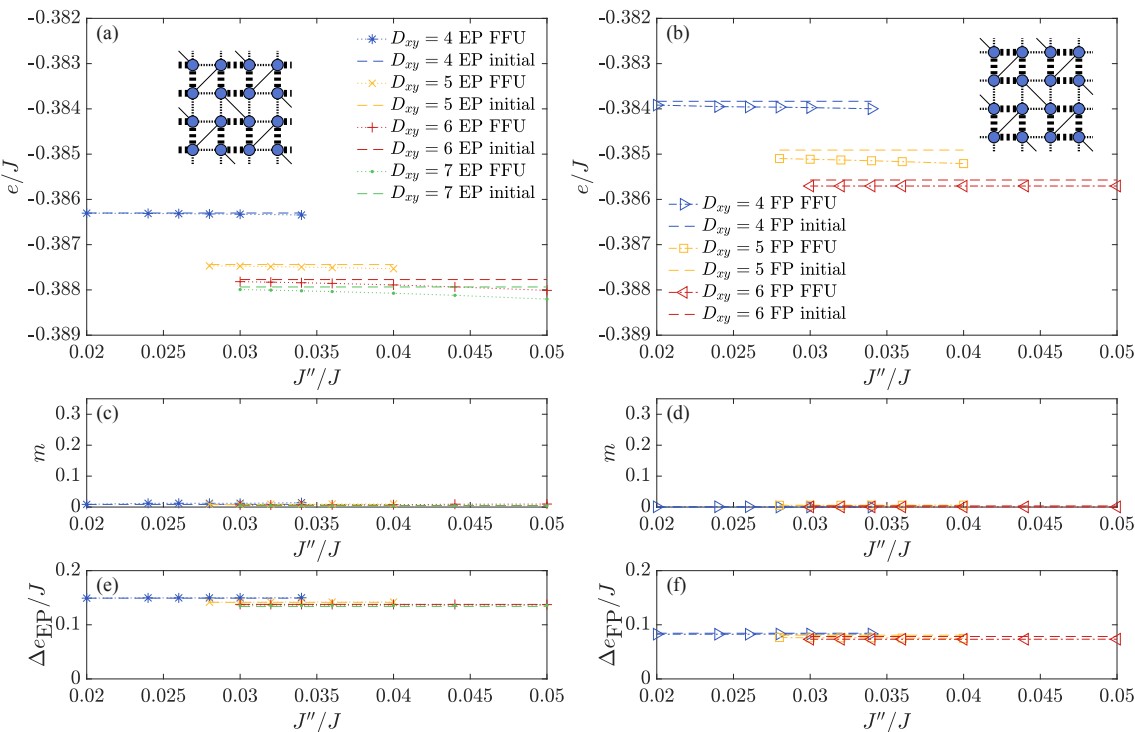

Figure 7: Comparison of results obtained for the EP (left column) and FP (right column) states for $J'/J = 0.7$ as a function of $J''/J$. (a)-(b) Energy per site of the two states for different values of $D_{xy}$ with $D_z = 3$. The dashed lines indicate the energies of the initial 2D states for $J''/J = 0$. (c)-(d) show the corresponding local magnetic moments, and (e)-(f) the EP and FP order parameters, respectively.

gain in interlayer energy is small in both cases. Thus, from these results we conclude that the interlayer coupling does not stabilize the FP state.

For completeness we also present the corresponding local magnetic moment and EP and FP order parameters in Figs. 7(c)-(f), where the latter is defined in a similar way as the EP order parameter (i.e., $\Delta e_{FP} = \bar{e}_{\text{other}} - \bar{e}_{FP}$). In both states the magnetization remains vanishingly small, and the strength of the corresponding plaquette orders does not change significantly with increasing $J'/J$.

## 3.4 Phase diagram for different bond dimensions

To get more insights into the finite bond dimension effects on the phase diagram, we present results for different $D_{xy}$ and $D_z$ in Figs. 8(a)-(b), respectively. Overall, on the scale of the phase diagram, the changes on the phase transition lines are hardly visible at large bond dimensions. As discussed in Sec. 3.1, the transition line between the dimer and Néel phase shifts to slightly lower values with increasing $D_{xy}$ and $D_z$ since the energy of the exact dimer state is fixed, whereas the Néel state energy gets lowered with increasing bond dimension. The EP to Néel transition line is initially shifted to larger values of $J''/J$ with increasing $D_{xy}$, but for $D_{xy} = 7$ essentially the same (slightly lower) value is found as for $D_{xy} = 6$. Increasing $D_z$ has the opposite effect, because the gain in energy for the Néel state is considerably higher than for the EP state, as we have seen in Sec. 3.2, but the change from $D_z = 2$ to $D_z = 3$ is very small. We note that since the EP state energy is only very weakly dependent on $J''/J$ and the dimer energy is constant, we have approximated the dimer to EP phase transition by a vertical line with the corresponding error bar from the 2D result [34].

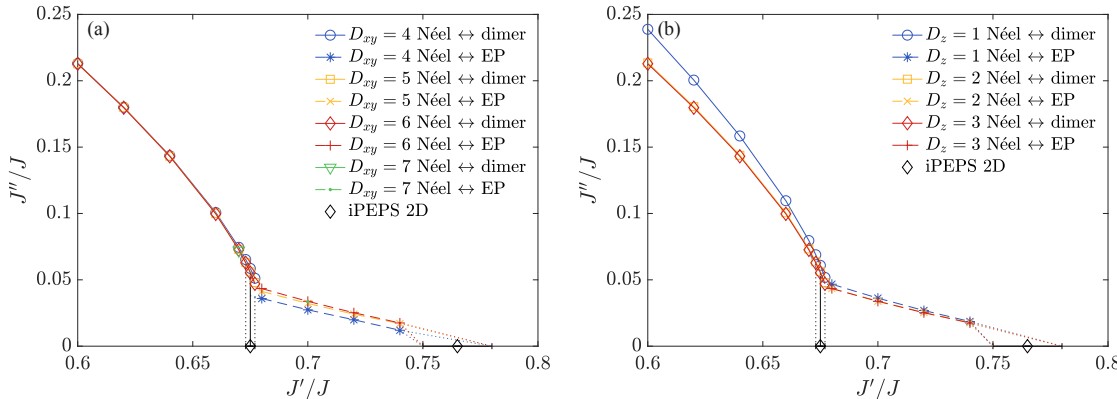

Figure 8: The phase diagram for (a) $D_{xy} = 4 - 7$ and fixed $D_z = 3$, and (b) fixed $D_{xy} = 6$ and $D_z = 1 - 3$. The phase transitions for the 2D model from Ref. [34] are shown by the black diamonds.

### 3.5 Estimate of the effective interlayer coupling in SrCu$_2$(BO$_3$)$_2$

Based on our phase diagram we can now discuss how large the effective value of $J''/J$ in SrCu$_2$(BO$_3$)$_2$ should be, such that the extent of the plaquette phase is compatible with experiments. It is clear that it cannot be beyond $J''/J \sim 0.05$, because at these values the plaquette phase vanishes entirely. To relate the pressure to the coupling ratio $J'/J$ we use the pressure model from Ref. [28], with the pressure dependence of the coupling parameters given by $J(p) = -5.13\,[\text{K/GPa}]\,p + 81.5\,[\text{K}]$ and $J'(p) = -1.43\,[\text{K/GPa}]\,p + 51.35\,[\text{K}]$. Based on specific heat measurements in Ref. [26] the phase transition between the plaquette and the Néel phase was found to occur around 2.5-3 GPa, corresponding to $J'/J = 0.704(8)$ and indicated by the light-red area in Fig. 4. At this coupling ratio the critical interlayer coupling in our phase diagram for ($D_{xy} = 6$, $D_z = 3$) is $J''/J = 0.032(4)$. Assuming that the pressure dependence of $J''$ is negligible compared to the pressure dependence of $J$, we find the effective interlayer coupling at ambient pressure to be around $J'' = 2.2(3)$ K or $J''/J = 0.027(4)$.

This value is compatible with DFT calculations [38] predicting $J''/J \lesssim 0.025$. However, it is substantially smaller than the estimates from fits to the magnetic susceptibility ($J''/J = 0.09$) [43] based on exact diagonalization of a 16-site system, which may be due to finite size effects, limitations of the mean-field ansatz which was used to include the interlayer effects, or the chosen temperature range which excluded the data below 100 K. We note that the LCTM approach can also be extended to finite temperature, offering the possibility to perform accurate fits to the magnetic susceptibility as done for the 2D model in Ref. [50], which is an interesting direction for future research. An even larger value was obtained in Ref. [44], $J''/J = 0.21$, based on an analysis of the bound state energies of the two-triplet excitations, however, the relevant coupling ratio was predicted to be $J'/J = 0.603$ and the phase transition was found at $J'/J = 0.63$ which is too low compared to more recent estimates.

Finally, one may wonder how this estimate would change in a slightly deformed Shastry-Sutherland model in which the FP state is slightly lower in energy than the EP state [40]. Since the change in energy of the AF state between $J''/J = 0$ and $J''/J = 0.1$ is roughly an order of magnitude larger than the typical energy difference between the EP and FP states, we can expect that the energies of the AF state and FP state intersect at a critical value $(J''/J)_c$ which is only slightly shifted compared to the standard SSM, yielding an estimate for the interlayer coupling that is not substantially different from the one we obtain for the standard SSM. A quantitative analysis would require simulations of the deformed model; however, the values of the effective couplings are not known.

## 4 Conclusion

In this work we have performed a systematic study of the phase diagram of the Shastry-Sutherland model with weak interlayer coupling using 3D iPEPS with the recently developed LCTM approach. Our results are in qualitative agreement with fourth-order SE results, however, on the quantitative level we found that the transition between the EP and the Néel phase occurs at a lower $J''/J$ than predicted by SE, with critical couplings of at most $J''/J \sim 0.04-0.05$. Based on our phase diagram and the extent of the plaquette phase found in experiments, we estimated the effective interlayer coupling in $SrCu_2(BO_3)_2$ to be around $J''/J \sim 0.027$ at ambient pressure. We also investigated the effect of the interlayer coupling on the competition between EP and FP phases and found that the dependence on $J''$ is very weak for both states, such that the EP state remains lower in energy than the FP state.

From the perspective of tensor network methods, applications to three-dimensional quantum systems have been very limited so far and they form an exciting and challenging frontier. Our work constitutes the first application of the LCTM approach beyond benchmark calculations, demonstrating the potential of iPEPS to explore challenging problems in the field of weakly-coupled layered systems.

## Acknowledgements

We acknowledge useful discussions with F. Mila. This project has received funding from the European Research Council (ERC) under the European Union's Horizon 2020 research and innovation programme (grant agreement Nos. 677061 and 101001604). This work is part of the D-ITP consortium, a program of the Netherlands Organization for Scientific Research (NWO) that is funded by the Dutch Ministry of Education, Culture, and Science (OCW).

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
