# Peer review of "Tensor network study of the Shastry-Sutherland model with weak interlayer coupling"

_SciPost Physics_

## Round 1 · Referee Report · Anonymous · 2023-5-4

Strengths
1- One of the first state-of-the-art calculation for a 3d frustrated quantum magnet
2- This paper provides an estimate of the weak interplane coupling for one of the most studied quantum material
Weaknesses
1- In order to compute the phase diagram, the authors compare energies of various phases. This is done using 1/D extrapolations (where D is the control parameter, i.e. the tensor bond dimension). However these extrapolations do not appear very reliable (the scaling is far from linear).
2- More importantly, the nature of the intermediate phase (full plaquette vs empty plaquette) seems to disagree with current experiments, so that the microscopic model is probably not valid ?
Report
This paper provides one of first tensor-network application to 3d frustrated quantum magnets. The team has a well-known expertise on this material and has already contributed to several studies of the 2d model. Recently, they have proposed an elegant algorithm to tackle a weak 3d coupling, which has been benchmarked on simpler models (where stochastic quantum Monte-Carlo simulations are possible). Hence, it is very interesting to see a more realistic simulation of this 3d model, especially given that other methods have tried in the past to estimate the 3d coupling (for instance series expansion).
Although the 3d model might not be fully realistic for the material (see weakness 2), it is interesting in its own. This paper is nicely written and provides compelling evidence that the 3d coupling is much smaller than previously anticipated.
Requested changes
1- How could the authors explain the discrepancy with susceptibility fits ? Isn't it related to the fact that the microscopic model is not adequate ? I would suggest to add a discussion about the modelling in the introduction.
2- Would it be possible to perform a finite correlation length scaling, especially in the Néel phase, to see if it improves the extrapolated value ?
Anonymous on 2023-04-19 [id 3598]
The paper explores ground state properties of the three-dimensional Shastry-Sutherland model. By means of the Tensor network method recently developed, the authors determine the phase diagram correctly. Therefore, this work is, of course, suitable for publication.
Short comment:
The three-dimensional orthogonal dimer model has first been discussed by Ueda & Miyahara.
(Kazuo Ueda and Shin Miyahara 1999 J. Phys.: Condens. Matter 11 L175) Therefore, this paper should be cited.
Author: Patrick Vlaar on 2023-04-28 [id 3630]
(in reply to Anonymous Comment on 2023-04-19 [id 3598])Thank you for the positive assessment of our work and for pointing out the reference. We will include it in the revised version.

---

## Round 2 · Referee Report · Anonymous · 2023-6-19

Report

The authors have answered all my comments and improved the manuscript accordingly. Using a numerical approach tailored for layered materials, they obtain a quantitative phase diagram for a well-studied material and also an estimate of the interlayer spin exchange. All these results will be relevant for the field and hence I recommend publication.

---

## Round 2 · Author Response

--- Response to the Referee ---

We would like to thank the referee for the thorough assessment of our work and for the valuable questions and comments, which we address in detail below.

Referee: "In order to compute the phase diagram, the authors compare energies of various phases. This is done using 1/D extrapolations (where D is the control parameter, i.e. the tensor bond dimension). However these extrapolations do not appear very reliable (the scaling is far from linear).
Would it be possible to perform a finite correlation length scaling, especially in the Néel phase, to see if it improves the extrapolated value?"

We thank the referee for this suggestion. Finite correlation length scaling (FCLS) would indeed enable a more accurate extrapolation of the energy and order parameters, but it can only be used in gapless systems with a diverging correlation length, i.e., the extrapolation would only be useful for the Néel phase, but not for the gapped plaquette phase. Furthermore, FCLS requires an accurate estimation of the correlation length, which, at least with the current version of the LCTM approach, is difficult to obtain in a controlled way since the approach is tailored to the computation of local quantities. Also, FCLS should ideally be combined with highly optimized states based on energy minimization, which has not been tested and implemented yet in the context of layered systems. Thus with the current methodology, unfortunately, we cannot perform an FCLS analysis. We have added a comment to mention this point in the paper.

We would like to stress, however, that we do not use the linearly extrapolated energy in 1/D as an estimate in the infinite D limit because the energy converges faster than linearly, so a linear extrapolation yields a value that is too low. Instead, we take the linear extrapolation as a lower bound for the energy, and the mean value of the lower bound and highest D value (upper bound) as the estimate, with an error bar given by half of the difference between the estimate and the lower bound (these types of estimates have been used already in several previous studies). Thanks to the fact that we can push the calculations to relatively large bond dimensions, this leads to relatively small error bars, typically smaller than 0.001J, which already provides a remarkably high accuracy for this challenging problem. We further note that the finite bond dimension effects on the phase boundaries are found to be very small, as can be seen in Fig. 8.

In the revised version we have rephrased the details on the extrapolation in Sec. 3.1 to be more clear, and we also explicitly refer to the extrapolation details in the Figure captions of Fig. 5 and Fig. 6.

Referee: “More importantly, the nature of the intermediate phase (full plaquette vs empty plaquette) seems to disagree with current experiments, so that the microscopic model is probably not valid?”

We thank the referee for raising this point which we would like to clarify in the following. First, we would like to stress that the Shastry-Sutherland model (SSM) has proven to be an excellent starting point to understand many properties of SCBO, not only qualitatively, but to a large extent also quantitatively, so in our opinion it is the most natural starting point to investigate the effects of the interlayer coupling.

It is true that there are indications from NMR and neutron scattering experiments that the intermediate phase is more compatible with the FP phase than the EP phase (as also mentioned in our paper). While additional experiments to confirm the nature of the intermediate phase would be highly desirable, we would like to stress that in the basic SSM, the EP and FP states are very close in energy, and the FP state becomes the ground state when considering a slightly deformed SSM (see Ref. [40]) with four different coupling parameters J_1, J_1’, J_2, J_2’. The precise values of these coupling parameters are not known, however. But even if the starting point would be a slightly deformed SSM, we do not expect a major change in our result of the effective interlayer coupling. The reason is that the FP and EP states are very close in energy with an energy difference of the order 0.001-0.002 J (and their energies are nearly independent of J’’), while the change in energy of the Néel state between J’’/J =0 and J’’/J=0.1 is roughly an order of magnitude larger. Thus, in a slightly deformed model in which the FP state is marginally lower in energy than the EP state, we can expect a transition value that is not substantially different from the one we find for the basic SSM.

We have added a discussion in Sec. 3.5 in the revised version to clarify this point.

Referee: “How could the authors explain the discrepancy with susceptibility fits? Isn't it related to the fact that the microscopic model is not adequate? I would suggest to add a discussion about the modelling in the introduction?”

This is an interesting question, and there may be several reasons for the discrepancy. First, we would like to stress that other estimates of the interlayer coupling have been obtained, besides the one based on susceptibility fits. In particular, the value obtained from DFT calculations in Ref. [38] is compatible with our result. Second, as argued above, we do not expect a major change of our estimate in a slightly deformed SSM, thus we believe this is not the reason for the discrepancy.

The fits to the magnetic susceptibility in Ref. [43] were based on a fitting range at high temperatures between 100 K and 350 K, based on exact diagonalization results of relatively small systems N_s=16, and using a mean-field type ansatz to include interlayer effects. Thus, the accuracy of the estimate may be limited due to the systematic errors of the mean-field ansatz, the restriction of the fitting range to high temperatures, and finite size effects (which would come into play particularly when including data also at lower temperatures). While a systematic study of the finite-temperature susceptibility is beyond the scope of this paper, we note that the LCTM approach could also be extended to finite temperature. This would enable us to systematically study the validity and limitations of the mean-field ansatz used in Ref. [43] and offer the possibility to make fits over more extended temperature ranges, using data at large bond dimensions, which yields considerably more accurate results than the 16-site finite size data, particularly around the specific heat peak (see the comparisons between iPEPS and exact diagonalization in Ref. [50] for the 2D model).

We have substantially extended the discussion about the comparison with previous results to mention the above points in Sec. 3.5 (which we found to be a better place to discuss these details than in the introduction), and we added a comment in the introduction to stress that the relevant couplings of the deformed model are not known.

---

## Round 2 · List of Changes

List of changes:
- We refined the discussion on the experimental observation of the Néel phase on p. 2.
- We added Ref. [29].
- We added Ref. [42] on p. 2.
- We added a comment about finite correlation length scaling in Sec. 3.1 and rephrased the details on the extrapolation to be more clear. We also explicitly refer to the extrapolation details in the Figure captions of Fig. 5 and Fig. 6.
- We have extended the discussion on the comparison between our estimate of the interlayer coupling with previous results in Sec. 3.5.
- We added a discussion about the change of the estimate of the interlayer coupling in a slightly deformed model (with full plaquette ground state) at the end of Sec. 3.5.
- We have recreated the figures and adapted the values in the text based on the refined LCTM contraction (using more iterations to compute the projectors).

---

## Editorial Decision

accepted_in_target_journal